# Effect of Alkaline Artificial Seawater Environment on the Corrosion Behaviour of Duplex Stainless Steel 2205

**Tien Tran Thi Thuy [1]** , **Krishnan Kannoorpatti [1,*]** , **Anna Padovan [2] and Suresh Thennadil [1]**

1 Energy and Resources Institute, College of Engineering, Information Technology and Environment, Charles Darwin University, Darwin, NT 0909, Australia; tien.tranthithuy@cdu.edu.au (T.T.T.T.); Suresh.Thennadil@cdu.edu.au (S.T.)
2 Research Institute for the Environment and Livelihoods, College of Engineering, Information Technology and Environment, Charles Darwin University, Darwin, NT 0909, Australia; Anna.Padovan@cdu.edu.au
* Correspondence: krishnan.kannoorpatti@cdu.edu.au



**Featured Application: A study about corrosion behaviour of duplex stainless steel in alkaline marine environment containing sulphate reducing bacteria.**

**Abstract:** Sulphate reducing bacteria (SRB) can be found in alkaline environments. Due to their metabolite products such as hydrogen sulphide, the corrosion behaviour of materials in alkaline environments may be affected by the presence of SRB. This study focuses on the investigation of corrosion behaviour of duplex stainless steel DSS 2205 in nutrient rich artificial seawater containing SRB species, *Desulfovibrio vulgaris*, at different alkaline conditions with pH range from 7 to 10. The open circuit potential value (OCP), sulphide level and pH were recorded daily. Confocal laser scanning microscopy (CLSM) was used to study the adhesion of SRB on the DSS 2205 surface. Electrochemical impedance spectroscopy (EIS) was used to study the properties of the biofilm. Potentiodynamic polarization was used to study the corrosion behaviour of material. Inductively coupled plasma mass was used to measure the concentration of cations Fe, Ni, Mo, Mn in the experimental solution after 28 days. Scanning electron microscopy (SEM) and energy dispersive X-ray spectroscopy (EDS) were used for surface analysis. The results showed that *D. vulgaris* are active in an alkaline environment with pH 7–9. However, at pH 10, *D. vulgaris* activity exhibited an 8-day lag. The corrosion rate of DSS 2205 at pH 9 was higher than at other pH environments due to a higher dissolved concentration of hydrogen sulphide.

**Keywords:** microbiologically influenced corrosion; sulphate reducing bacteria; seawater; duplex stainless steel

## 1. Introduction

Microbiologically influenced corrosion (MIC) is a serious problem in various industries [1–3]. MIC can happen not only in oxic environments but also in anoxic environments (e.g., marine sediments, deep seawater, water-logged soil) where anaerobic bacteria occur. Sulphate reducing bacteria (SRB) were found to be one of the predominant types of bacteria associated with MIC [4,5]. In the absence of oxygen or under low oxygen conditions, these bacteria catalyse sulphate ions to sulphides and corrode metals through a series of oxidation and reduction reactions [1,6]. Corrosion of stainless steel is primarily due to the presence of sulphide ions [7,8] which can form metal sulphides and deteriorate the passive film on the metal [4,6,9].

Duplex stainless steel, a well-known highly corrosion resistant material, was reported to be more corrosion resistant at higher pH values than at lower pH in abiotic environments [10] though a tenacious passive film of chromium. Alkaline materials (e.g., carbonates) can combine with calcium and magnesium from the environment to form scale at high pH which may help to protect stainless steel from general corrosion [11]. However, it is not clear how duplex stainless steel with its high chromium content and passive film will be affected at high pH and in the presence of SRB.

At present, most of the research focusses on studying SRB—induced corrosion at a specific pH, commonly the optimum pH for bacteria to grow [2,9,12–22] which is about 7.4. However, pH of environments can play an important role in microbial corrosion [4,23]. The bacterial metabolite products such as sulphide ions can accelerate corrosion processes as sulphide is aggressive to passive film on stainless steel [24–26]. Sulphide enhances the susceptibility of materials to corrosion through different ways such as cathodic depolarisation [27], changes to the local pH upon hydrolysis reactions of dissolved metal ions and initiates pitting [28], supporting active dissolution of steel [29], and increasing electron uptake through metal sulphides [30]. Previous studies showed that at higher pH values, the dissolution rate of $H_2S$ is faster [31] which increases the concentration of sulphide ions resulting in an increased corrosion rate of carbon steel materials [32].

Environmental pH levels are an important factor for microbial corrosion by SRB but only few details were given [6,33,34]. However, microbial corrosion in alkaline environments has received less attention as most microbes are not active in this environment [35]. SRB were found to be inhibited at pH above 9 [36] but can tolerate up to pH 9.5 [37,38]. SRB have even been found in environments at pH 10 [39–42] which has implications for equipment operating in alkaline environments such as soda lakes, mine water discharges [41], and paper mills [42]. For instance, nuclear waste duplex stainless steel containers in concrete bunkers can be subjected to MIC from high pH water percolating through the concrete [35]. The presence of SRB in alkaline environments can accelerate the corrosion of equipment operating in such conditions, resulting in equipment failure.

An understanding of microbial corrosion processes under alkaline conditions would help to find suitable methods for corrosion control. However, there is not much detailed information in the literature which addresses the influence of environmental alkaline pH on microbial corrosion of materials or associated mechanisms. Many studies do not measure the pH during the corrosion process and the growth of bacteria. Although SRB was reported to survive in alkaline solutions, it is unclear as how they survive these environments and how they interact with metals. This paper tries to answer some of these questions. Even a recent review by Little et al. [4] has not discussed the effect of high pH on microbes' growth and their effect on microbial corrosion. The present work is aimed to study the corrosion behaviour of duplex stainless steel in an alkaline seawater environment in the presence of sulphate reducing bacteria in order to gain better understanding of the corrosion mechanism caused by SRB. Electrochemical characteristics, biochemical and surface analyses were performed to reveal the corrosion behaviour of the duplex stainless steel in alkaline pH environments. A possible mechanism of microbial corrosion process of materials and the change in environmental pH is proposed.

## 2. Materials and Methods

### 2.1. Materials

Four duplex stainless steel 2205 (DSS 2205) coupons (10 mm × 10 mm × 2 mm) were used for the experiments in 4 different pH environments: pH 7, pH 8, pH 9, and pH 10. They were mounted in a mould of non-conducting epoxy resin with a connected copper wire to act as a working electrode for electrochemical testing (Figure 1).

An additional four coupons of duplex stainless steel 2205 (10 mm × 10 mm × 2 mm) were prepared to study bacterial adhesion and biofilm formation. Three samples were immersed in each pH environment.

All the coupons were polished to around 1 μm finish. After polishing, the coupons were rinsed with water, degreased with acetone, rinsed with distilled water, immersed in 80% ethanol for 2 h and finally dried in a biohazard cabinet to prevent bacterial contamination before the experiments. The chemical composition of DSS 2205 determined through energy dispersive X-ray fluorescence spectrometry (EDS-8100, Shimadzu, Japan), is given in Table 1.

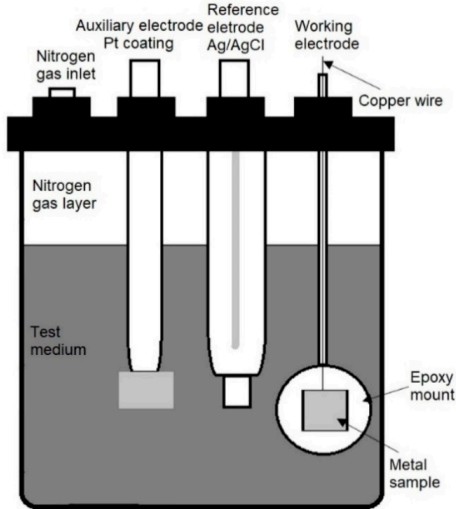

**Figure 1.** Experimental set up showing a three-electrode cell with nitrogen gas layer to maintain anaerobic test conditions.

**Table 1.** Chemical composition of DSS 2205.

| Duplex Stainless Steel 2205 | Chemical Elements (%) | | | | | | | | |
|---|---|---|---|---|---|---|---|---|---|
| | **Fe** | **Mn** | **S** | **V** | **Si** | **Cr** | **Ni** | **Cu** | **Mo** |
| | 66.32 | 1.68 | 0.05 | 0.11 | 0.43 | 22.1 | 6.12 | 0.30 | 2.89 |

*2.2. Medium and Test Conditions*

The studies were performed in nutrient rich artificial seawater. This consisted of modified Baar's medium (g L$^{-1}$): MgSO$_4$ 0.2; sodium citrate 0.5; CaSO$_4$ 0.1; NH$_4$Cl 0.1; K$_2$HPO$_4$ 0.05; sodium lactate 3.5; yeast extract 1.0 added to 4 L of artificial seawater prepared according to ASTM 114-98 (g L$^{-1}$) [43]: NaCl 24.53; MgCl$_2$ 5.2; Na$_2$SO$_4$ 4.09; CaCl$_2$ 1.16; KCl 0.695; NaHCO$_3$ 0.201; KBr 0.101; H$_3$BO$_3$ 0.027; SrCl$_2$ 0.0025, NaF 0.003 and high pure water.

The test medium was distributed to eight 500 mL glass bottles. The pH was adjusted to pH 7, 8, 9 and 10 using 1 M hydrochloric acid or 1 M sodium hydroxide. The test medium was purged with nitrogen gas for 1 h and sterilized by autoclaving for 15 min at 121 °C.

*Desulfovibrio vulgaris* (ATCC® 7757), a species of SRB, was used in this study (In Vitro Technologies, Victoria, Australia). The strain was stored at −80 °C in 15% glycerol. Bacteria were retrieved from stock and cultivated in 500 mL modified Baar's medium (Table 2) for 48 h at 37 °C under anaerobic conditions. After 48 h, 10 mL of bacteria culture medium was removed to determine bacterial concentration. The bacterial cells were harvested by centrifugation (5000 rpm, 10 min) and resuspended in 10 mL high pure water for enumerating bacteria cells following staining with 0.4% trypan blue and counting using a hemocytometer. Five mL of culture medium was added to each 200 mL glass bottle containing nutrient rich artificial seawater to give a final bacteria concentration of approximately 2.23 × 10$^3$ cells mL$^{-1}$ stock solution.

Additionally, phosphate-buffered saline 1X (PBS) and 4% formaldehyde in PBS solution were prepared for staining bacteria with 4′,6-diamidino-2-phenylindole (DAPI) and observing the biofilm by confocal laser scanning microscopy.

### 2.3. Analytical Methods

The experiments were carried out for 28 days at 37 °C which falls within the optimum temperature range for growth of mesophilic bacteria. The experiment included corrosion testing (electrochemical testing, measuring dissolved sulphide, pH variation, ICPMS and surface analysis) and adhesion testing. All experiments were repeated twice.

### 2.4. Biofilm Formation

Four coupons were placed in four 500 mL bottles containing nutrient rich artificial seawater at initial pH values of 7, 8, 9 and 10. A ten mL of bacteria stock solution was transferred aseptically to these bottles of medium and incubated at 37 °C for 28 days ensuring the coupons were fully immersed in the test medium with the polished sides facing upward. The coupons in each bottle were removed after 2 days of immersion, gently rinsed 3 times with 1X PBS to remove poorly adhered bacteria, and then fixed in 350 μL of 4% formaldehyde for 20 min. Coupons were rinsed 3 times with 1X PBS, stained with 350 μL of 300 nM DAPI in PBS solution and incubated in the dark for 5 min. The coupons were rinsed 3 times with 1X PBS before microscopic observation using confocal laser scanning microscope (CLSM) (ZEISS LSM 510 META, Zeiss, Germany).

### 2.5. Electrochemical Testing

All electrochemical experiments were performed in a three-electrode cell (Figure 1). A platinum coated electrode was used as counter electrode, an Ag/AgCl electrode was used as a reference electrode. A nitrogen gas layer was added to the top of the cell to create fully anaerobic conditions inside the cell. The electrochemical experiments were performed using potentiostat (VERSASTAT3-300, AMETEKSI, Berwyn, PA, USA, 2018) and the results analysed using VersaStudio software. Open circuit potential (OCP) value of each specimen was recorded daily.

### 2.6. EIS (Electrochemical Impedance Spectroscopy)

EIS was recorded after stabilization at 2 h, 2 days, 4 days, 14 days, 18 days, 22 days and 28 days. EIS has been used to study MIC, its biofilm formation and its interaction with material surface [44]. The tests were carried out at OCP and the amplitude value was 20 mV with frequency range from 0.05 Hz to 100,000 Hz. The impedance data was analysed by an equivalent circuit utilizing software ZsimpWin (AMETEKSI, Berwyn, PA, USA, 2018) which was integrated with VersaStudio (AMETEKSI, Berwyn, PA, USA, 2018).

### 2.7. Sulphide Level and pH Level

Samples from each bottle were taken daily using a sterile syringe. Sulphide levels were measured using a colorimeter (Hach DR300, HACH, Loveland, CO, USA, 2016). The pH of each sample was measured using a pH meter. Each measurement was repeated twice.

### 2.8. Inductively Coupled Plasma Mass Spectrometry (ICPMS) Analysis

On the final day of the experiment, 15 mL solution from each bottle was filtered through a 0.45 μm filter to remove bacteria. The solutions were then analysed for metal concentrations in an inductively coupled plasma mass spectrometer (ICPMS) (Agilent 7500ce ICPMS, Agilent, Santa Clara, CA, USA, 2015) which is an octopole reaction system using a standard addition calibration method for seawater. The reporting limit for Cr, Mn, Fe, Ni and Mo are 0.20, 0.20, 4.20, 0.10, 0.30 ppb, respectively. This experiment was repeated three times.

### 2.9. Surface Analysis

All the working electrodes were removed from the solution after 28 days, then washed gently three times with 1X PBS and fixed with 4.5% glutaraldehyde in PBS for 30 min. The glutaraldehyde

was removed, and the coupons were washed three times with ultrapure water. The coupons were dehydrated in an ethanol series (25, 50, 75, 90, and 100%) for 10 min each, dried in a biohazard cabinet for 30 min and mounted. Chemical analysis of the biofilm and/or corrosion products on the surface of the coupons was carried out using energy scanning electron microscope (SEM) (JEOL 5800LV, JEOL, Akishima, Tokyo, Japan, 2010) fitted with energy dispersive spectrometer (EDS) at cursor of 20 keV with full scale of 1100 counts. To observe the pitting formation on the surface of coupons, the biofilm and corrosion products were removed by washing three times with 100 mL high purity water, then cleaning in an ultrasonic water bath for 2 min. The coupons were then immersed in Clarke's solution according to ASTM standard G1-03 [45]. Finally, all coupons were rinsed in high purity water followed by 80% ethanol and dried in biohazard cabinet. The cleaned samples were analysed by SEM at 20 keV.

## 3. Results

### 3.1. Sulphide Concentration and pH Variation

The change in dissolved sulphide concentration at different starting pH values during the 28-day experiment is shown in Figure 2. The dissolved sulphide concentration increased immediately at pH 7, 8 and 9, reaching a maximum at around day 14, then remaining stable for few days before starting to decline. At pH 10, there was a lag of approximately 8 days before dissolved sulphide concentrations increased, reaching a peak of similar magnitude to the other pH experiments on day 22 before decreasing. This could indicate that for the first few days, there was little sulphide produced due to the high pH environment. A possible explanation for this is that the concentration of $OH^-$ in the environment was too high and out of range for bacteria to live as the suitable range of pH for *D. vulgaris* is 5–9. Therefore, during the first 8 days, a portion of bacteria died, or cells were shocked and unable to function properly, leading to an extended lag phase in bacterial growth. Sulphide level started to increase after 8 days indicate the growth of bacteria after phase lag.

Environmental pH of the bulk environment changed during exposure time (Figure 3). After approximately 14 days, the pH of all the solutions were changed to around 7.1, then gradually increased to around 7.5 after 28 days.

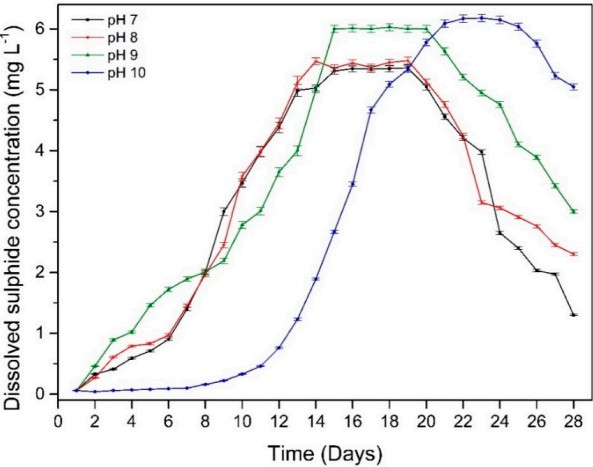

**Figure 2.** Dissolved sulphide levels over time for samples at different starting pH.

### 3.2. Biofilm Formation

The extent of bacterial adhesion and biofilm formation on the coupon surface after 2 days immersion as observed by CLSM is presented in Figure 4. Bacterial adhesion was most extensive at pH 9, with the least amount of adhesion at pH 10. Biofilm thickness was greatest at pH 7, 8 and 9, over 75 μm thick in some sections. In contrast, coupons immersed in the solution of initial pH 10 solution had thinner biofilm thickness with an average thickness of 15 μm.

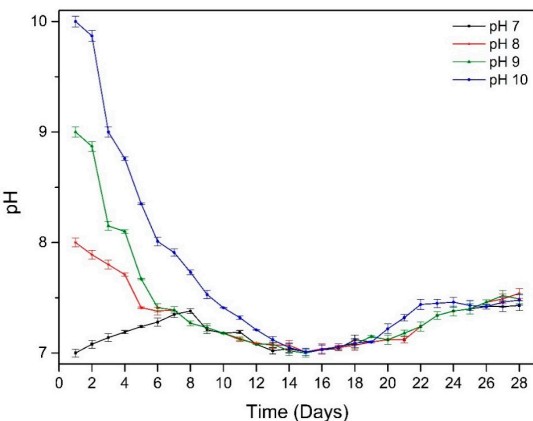

**Figure 3.** The change in pH over time for samples at different starting pH.

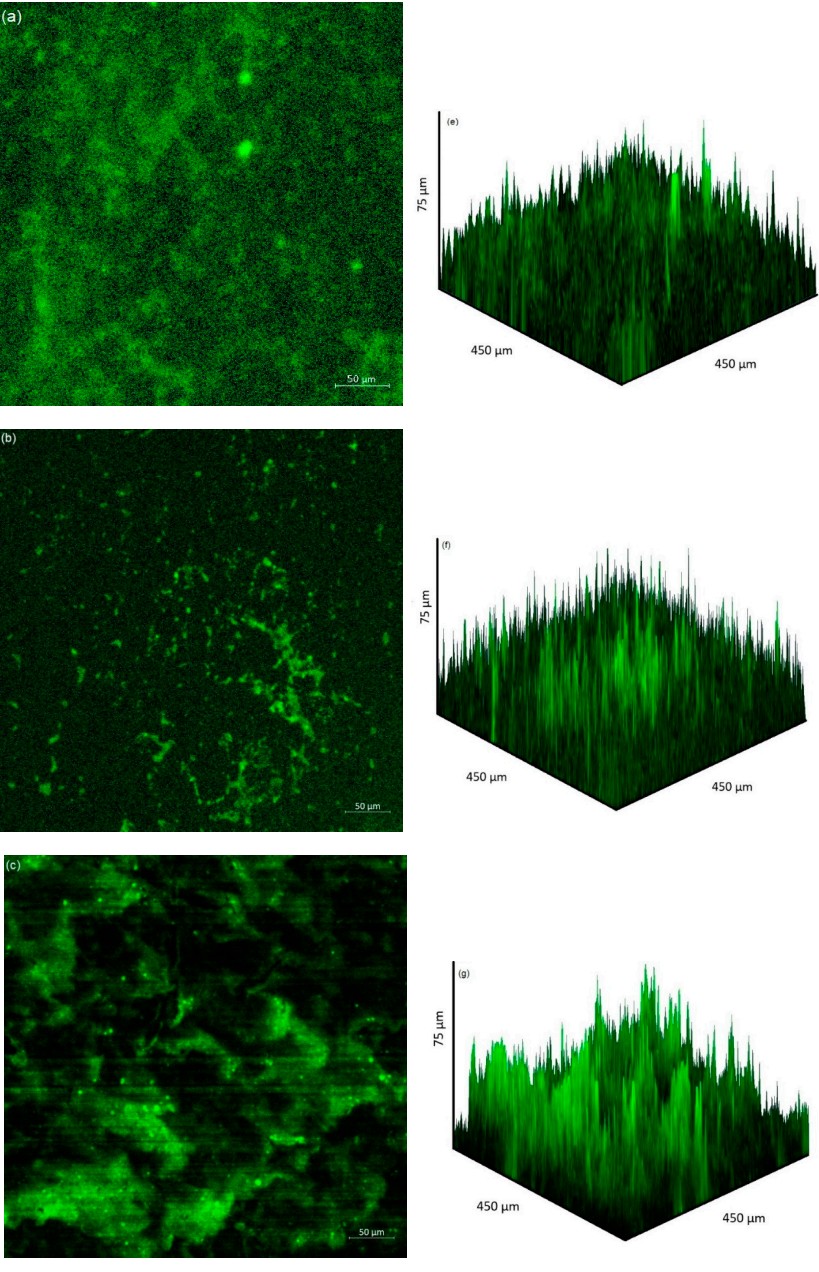

**Figure 4.** *Cont.*

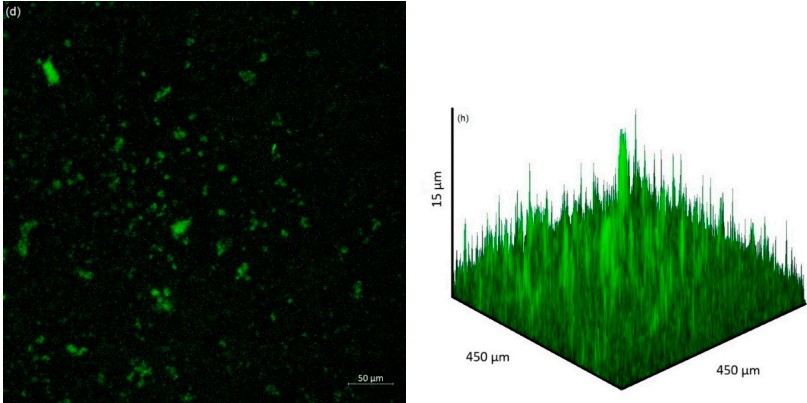

**Figure 4.** CLSM images of bacterial attachment (2D) and biofilm formation (3D) after 2 days immersion of coupons at pH 7 (**a**,**e**), (**b**) pH 8, (**b**,**f**) pH 9, (**c**,**g**) pH 10 (**d**,**h**).

### 3.3. Electrochemical Testing

The OCP value of the working electrode in different pH environments declined rapidly in the first 8 days due to the formation of biofilm on the surface of the working electrode making the electron transfer between the working electrode surface and solution more difficult (Figure 5). The OCP value then increased slightly in all pH experiments which indicates the beginning of corrosion of the samples.

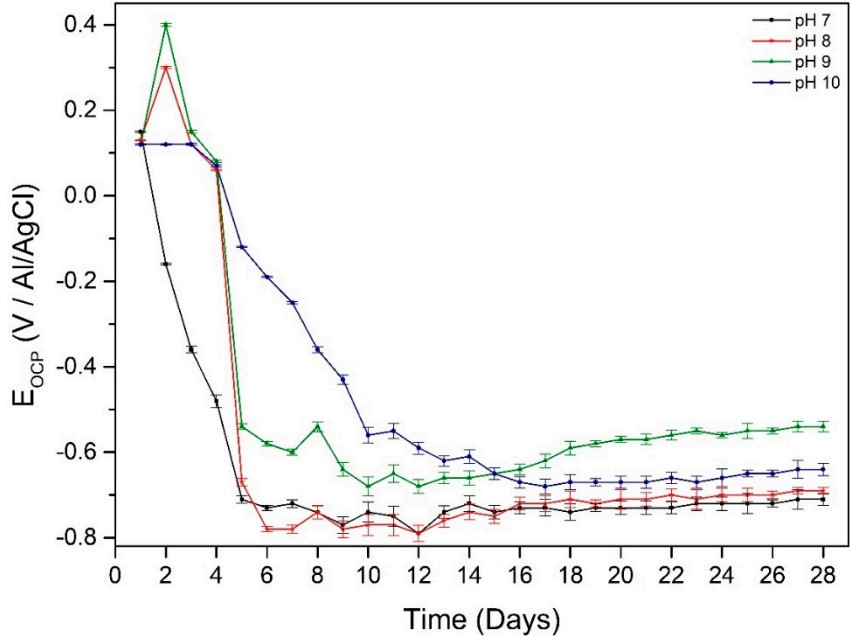

**Figure 5.** OCP curves of DSS 2205 in pH 7, 8, 9 and 10 environments over time for samples at different starting pH.

The EIS spectra of working electrodes immersed in different pH environments (7–10) after different exposure times is presented in Figure 6. The Nyquist plots reveal a capacitive arc representing the resistance of the film formed on the electrode surface. The film can be a passive film, biofilm or even corrosion product layer. The higher the radius of the capacitive arc, the higher the resistance of the electrode to corrosion. Different equivalent electric circuit (EEC) models have been proposed for interpreting impedance spectra of materials in microbial environment. The model $R_s[Q_{CPE}[R_b[C_{dl}R_{ct}]]]$ has been used for studying microbial corrosion [18,46]. Passive film along with biofilm formation on the surface of materials can act as a double layer capacitance [18,46]. In this research, this model was used to fit the experimental data as it can present the double layer capacitance. $R_s$ represents

the resistance of solution, $R_b$ is the resistance of biofilm/passive film formed on material surfaces, $R_{ct}$ is the charge transfer resistance, $C_{dl}$ is the capacitance of electrical double layer. A constant phase element (CPE) was introduced to the model as it presents a deviation from a true capacitive behaviour. It is interesting to note that after stabilization at open circuit potential for 2 h, the working electrode in pH 10 had the highest impedance as it has the highest radius of capacitive arc even though it had the lowest biofilm thickness on the surface.

The overall impedance of working electrodes increases with immersion time (0–14 days) and this reveals the thickening of protective film on the surface of the electrodes. This can be due to the biofilm on the surface acting as a barrier to protect the surface of the material. However, after 14 days, the impedance starts to decrease, especially at pH 9 which reveals that the protective ability was lost. This indicates the corrosion started occurring after 14 days exposure, and this is in good agreement with OCP variation (Figure 5). After 28 days exposure, the impedance of the working electrodes at different pH values changed dramatically compared to the first 2 days of exposure. The working electrode immersed in pH 9 solution had the lowest impedance after 28 days immersion indicating the thinnest protective film on the surface and therefore more susceptible to pitting. This can be seen in the OCP value after 28 days immersion (Figure 5) when the working electrode in pH 9 environment had the highest OCP value.

The total resistance ($R_{tot}$) was defined by the total resistance of biofilm/passive film layer formed on working electrodes surfaces and charge transfer resistance ($R_{tot} = R_b + R_{ct}$), where $R_b$ and $R_{ct}$ were parameters obtained from EIS fitting procedure. The total resistance of all electrodes increased rapidly, reaching a peak after 14 days at pH 7, 8 and 9, due to the formation of biofilm on the surface (Figure 7). The working electrode impedance at pH 10 reached a maximum after 18 days of immersion. This is longer than the time taken to reach the maximum impedance in the lower pH environments. This delay in reaching the maximum impedance can be attributed to the extended lag phase of bacterial growth. The total resistance of all electrodes declined sharply after reaching their peaks. This indicates that corrosion has started to occur. However, the drop rate of total resistance of all electrodes then decreased as can be evidenced from the decrease in the drop rate after a few days. This can reveal the total resistance this included the corrosion products layer.

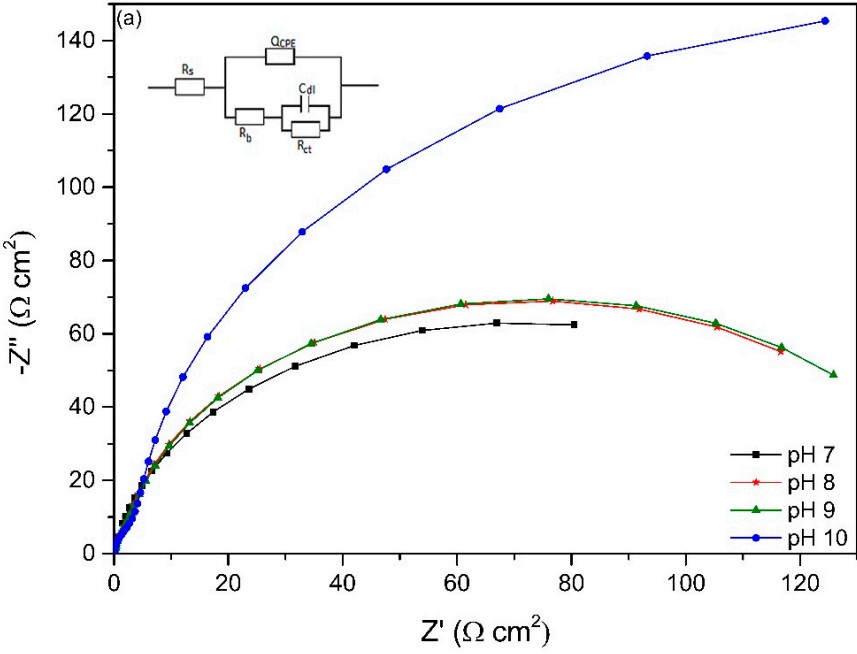

**Figure 6.** *Cont.*

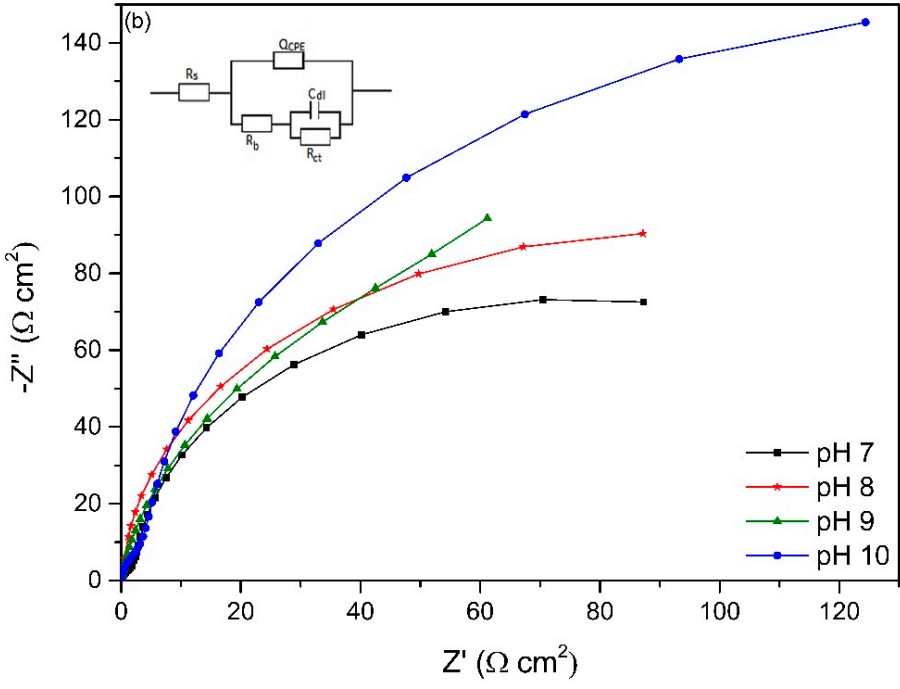

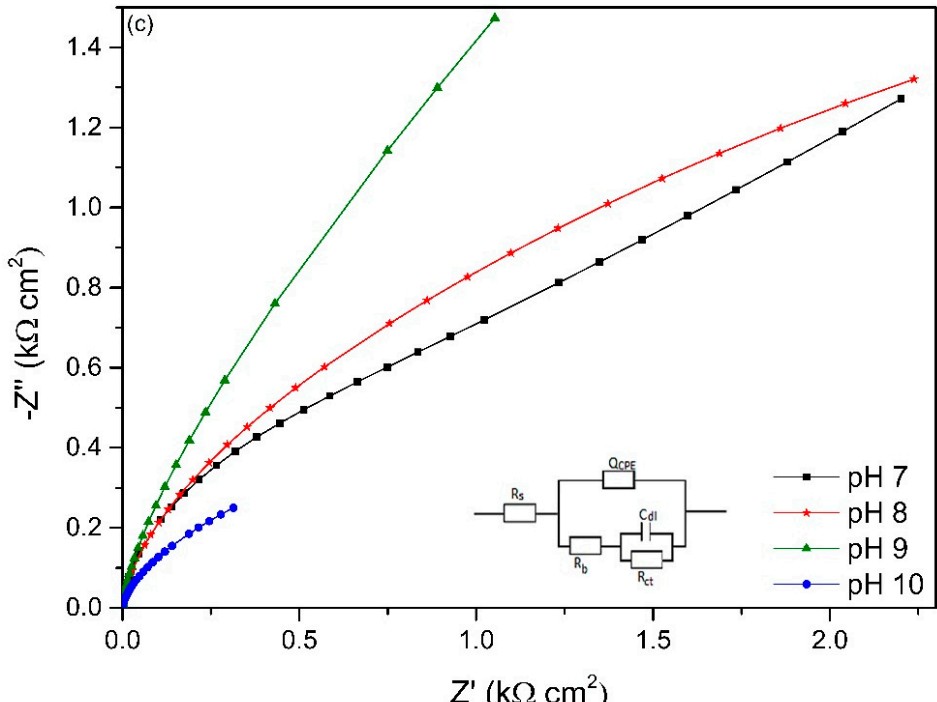

**Figure 6.** *Cont.*

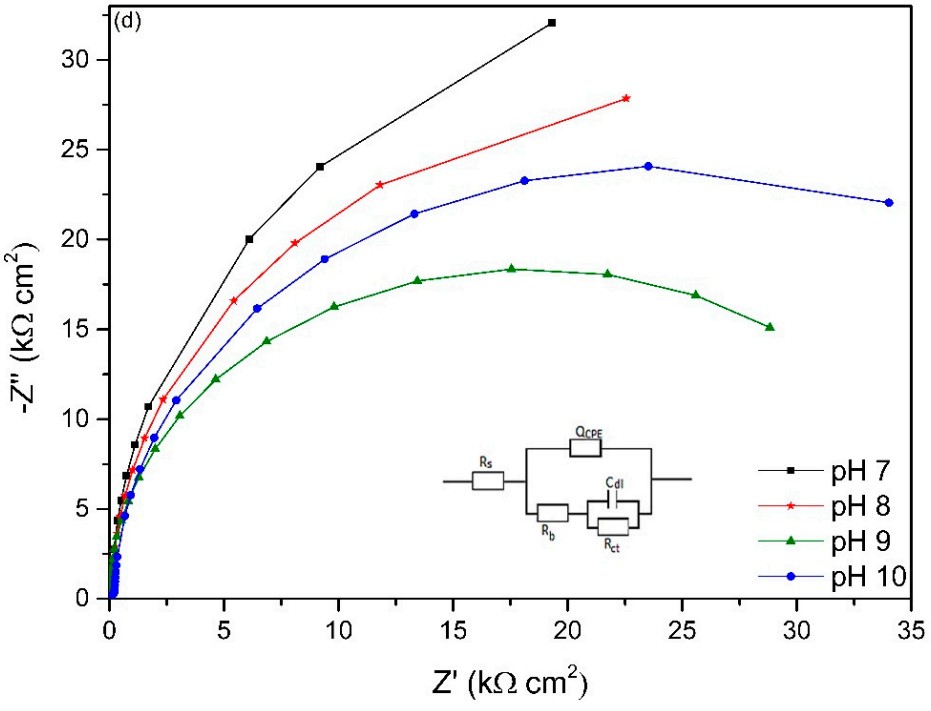

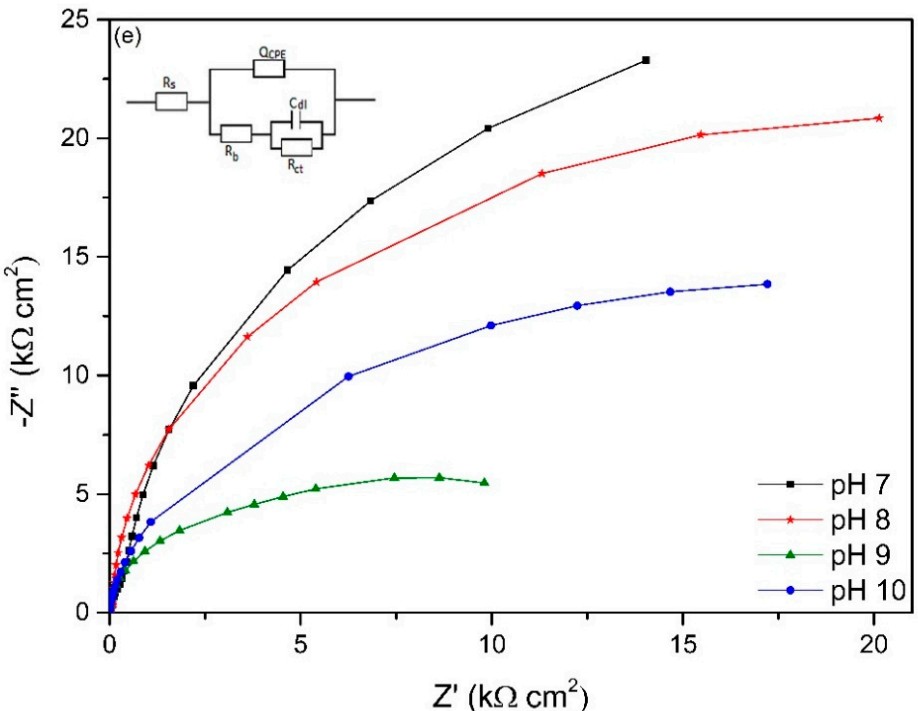

**Figure 6.** *Cont.*

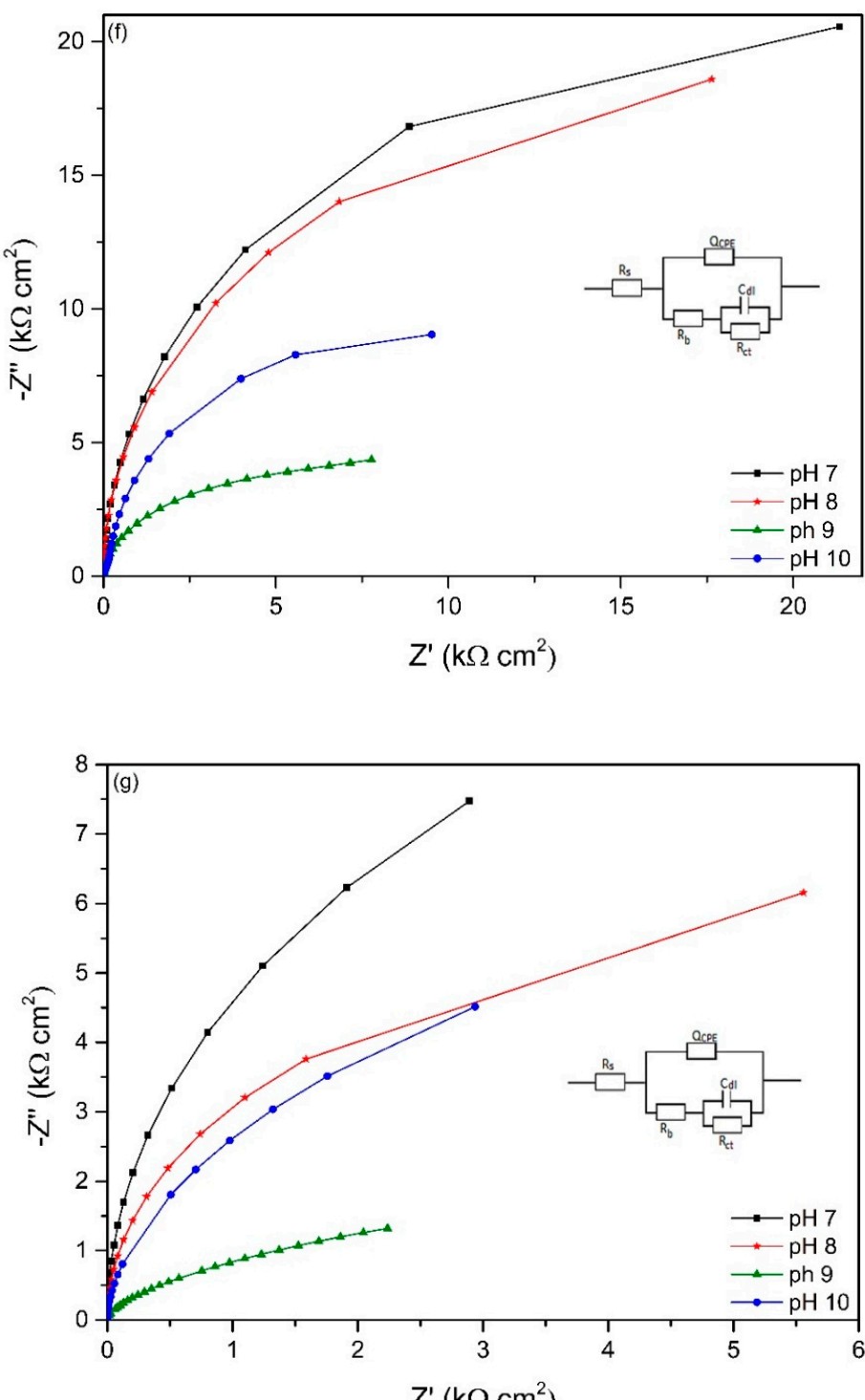

**Figure 6.** Nyquist plots of working electrodes at different starting pH after (**a**) 2 h, (**b**) 2 days, (**c**) 4 days, (**d**) 14 days, (**e**) 18 days, (**f**) 22 days and (**g**) 28 days exposure based on the EEC model in each figure.

The change in double layer capacitance ($\Delta C_{dl}$) during exposure time of working electrodes in different pH environments is shown in Figure 8. The change in double layer capacitance in this study was calculated by Equation (1) according to previous methods of studying bacterial adhesion by double layer capacitance [47]

$$\Delta C_{dl} = \frac{C_t - C_0}{C_0} \times 100\%$$

(1)

where $C_t$ and $C_0$ are the double layer capacitance at specific time of exposure and at initial time, respectively. These values were obtained from EIS measurements.

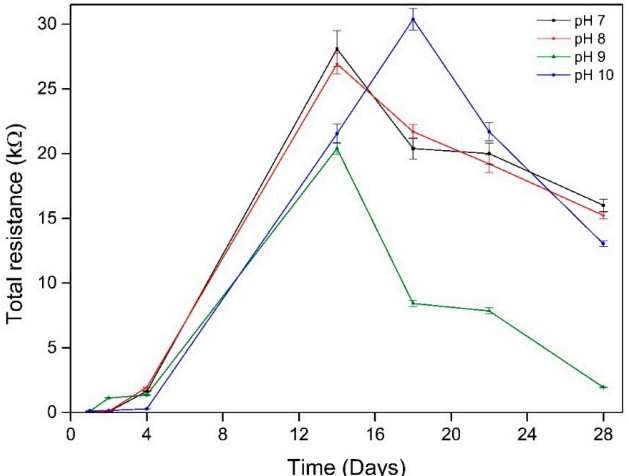

**Figure 7.** The evolution of total resistance of the working electrodes over time at different starting pH.

The amount of bacterial attachment can be correlated to the double layer capacitance [47] which is defined by Equation (2)

$$C_{dl} = \varepsilon \varepsilon_0 \frac{A}{d} \qquad (2)$$

where $\varepsilon$ is the dielectric constant of the electrolyte, $\varepsilon_0$ ($8.854 \times 10^{-12}$ F/m) is the permittivity of free space, $d$ (m) is the thickness of the double-layer (represent by biofilm), and $A$ (m$^2$) is the electrode area [47]. The higher the thickness of biofilm, the lower the double layer capacitance.

The change in double layer capacitance was correlated to the degree of bacteria adhesion and biofilm maturity [48]. The change in double layer capacitance ($\Delta C_{dl}$) of the working electrode immersed in pH 10 environment was around 0.2 for the first few days of exposure which could be explained by the bacterial growth remaining in the lag phase during this time. The decrease of $\Delta C_{dl}$ indicates the increase of biofilm formed on material surface which made the total resistance of working electrode to increase (Figure 8).

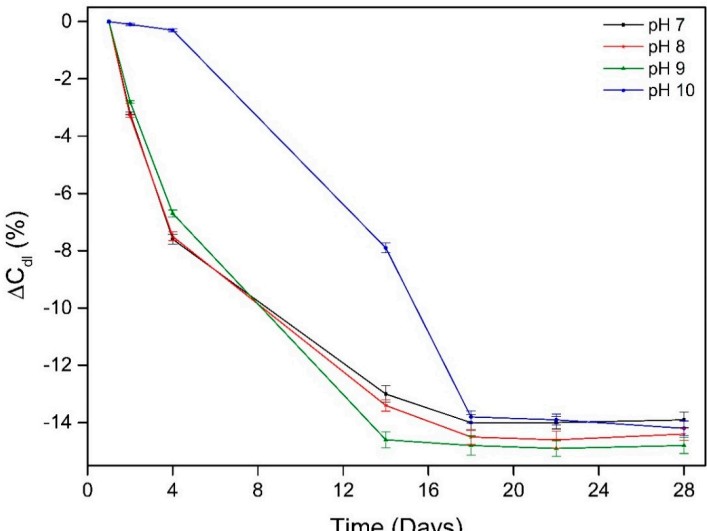

**Figure 8.** Change in double layer capacitance during exposure time of working electrodes in different pH environments.

Figure 9 presents the potentiodynamic polarization behaviour at different pH. The corrosion current densities and corrosion potentials are presented in Table 2. The working electrode immersed in pH 9 environment had the highest current density and therefore highest corrosion rate compared to samples at other pH values.

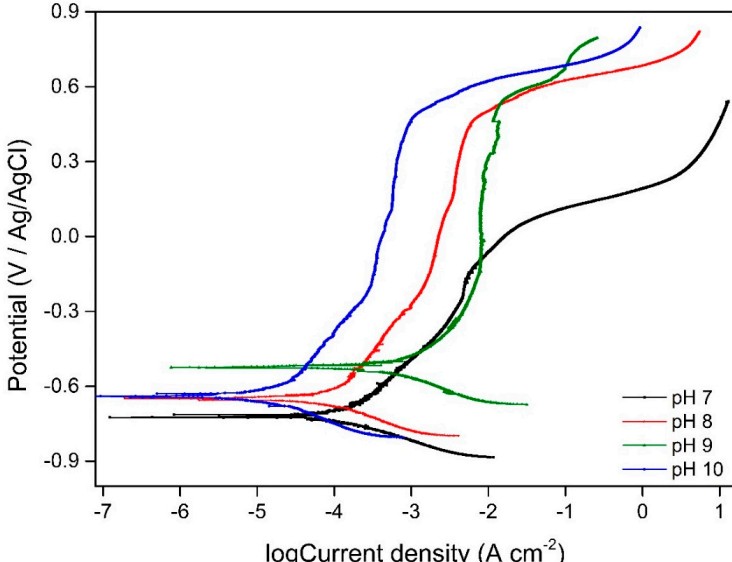

**Figure 9.** Potentiodynamic polarization curve of working electrode in different pH environments.

**Table 2.** Corrosion density and potential of working electrode in different pH environments.

|         | $I_{corr}$ (mA·cm$^{-2}$) | $E_{corr}$ (mV/Ag/AgCl) |
| ------- | ------------------------- | ----------------------- |
| pH 7    | 0.166                     | −707                    |
| pH 8    | 0.228                     | −685                    |
| pH 9    | 0.867                     | −542                    |
| pH 10   | 0.282                     | −641                    |

### 3.4. Metal Concentrations

The concentration of cations (Cr, Mn, Fe, Ni, Mo) in the media solution with different initial pH varied between samples are presented in Figure 10. The highest concentration of cations was measured for samples immersed in pH 9 solution indicating that the highest corrosion rate occurred in these samples which is in good agreement with potentiodynamic polarization results (Figure 9 and Table 2).

### 3.5. SEM Images of COUPONS immersed in Different pH Environments

The SEM images of coupons immersed in different pH environments after biofilm removal are presented in Figure 11. Pits were evident as dark circular marks on the surface of the coupons due to corrosion. The samples before the corrosion experiments were found to have no pits or voids (no photos available). Visual inspection revealed that the number and size of pits were greater in samples immersed in pH 9 than the others, indicating a higher corrosion rate in the pH 9 environment. This is in good agreement with both potentiodynamic polarization results (higher current density of coupons at pH 9) and ICPMS results (higher dissolved metal concentrations at pH 9). Element concentrations of the coupons are shown in Table 3. Sulphur was detected on the surface of the coupons in the matrix of corrosion products and biofilm and was highest in coupons immersed at pH 9. Sulphur generally is known to form metal sulphide as a corrosion product [15,33,49].

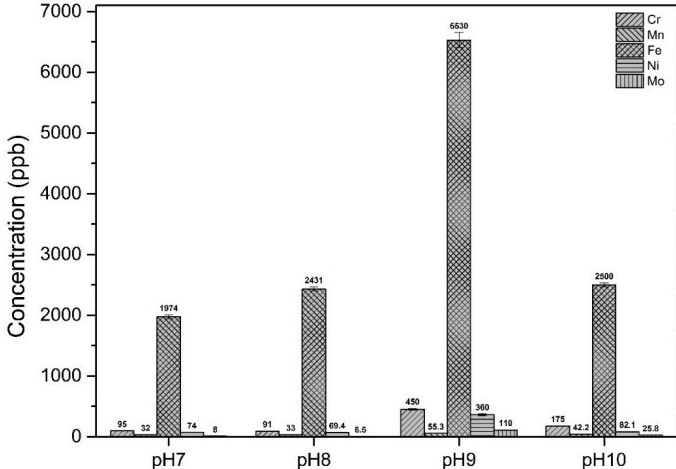

**Figure 10.** Concentrations of chromium, manganese, iron, nickel and molybdenum in the medium following immersion of coupons for 28 days in different pH environments.

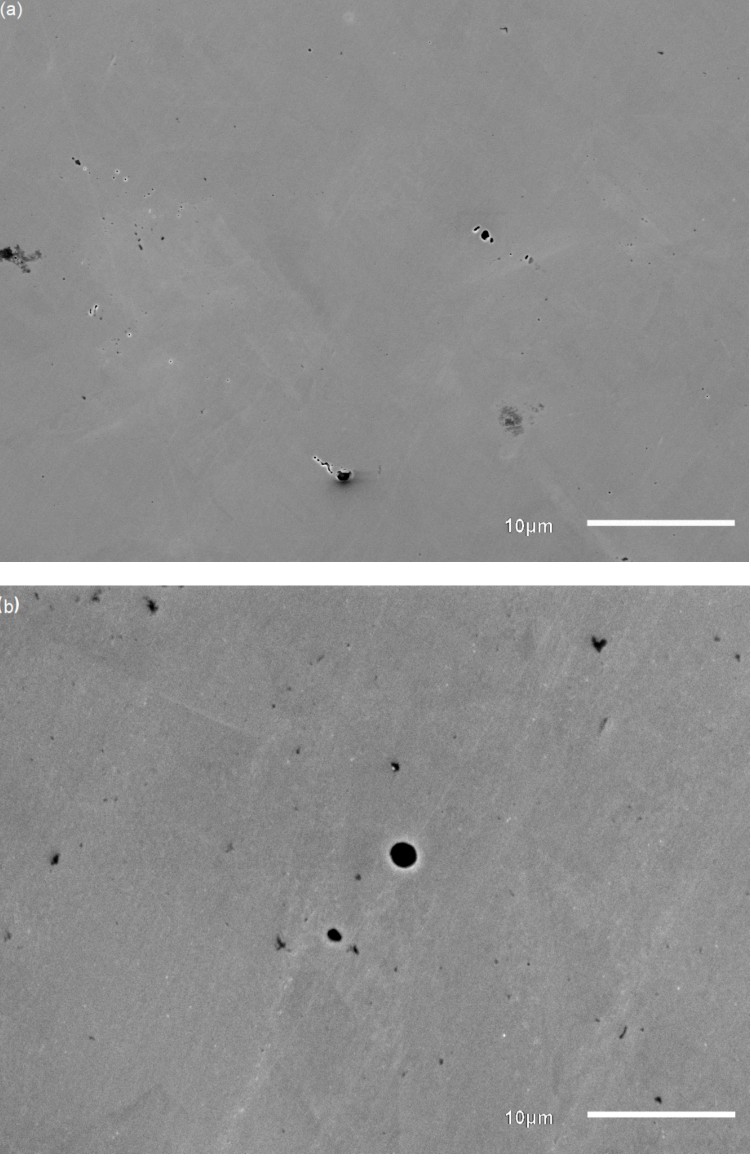

**Figure 11.** *Cont.*

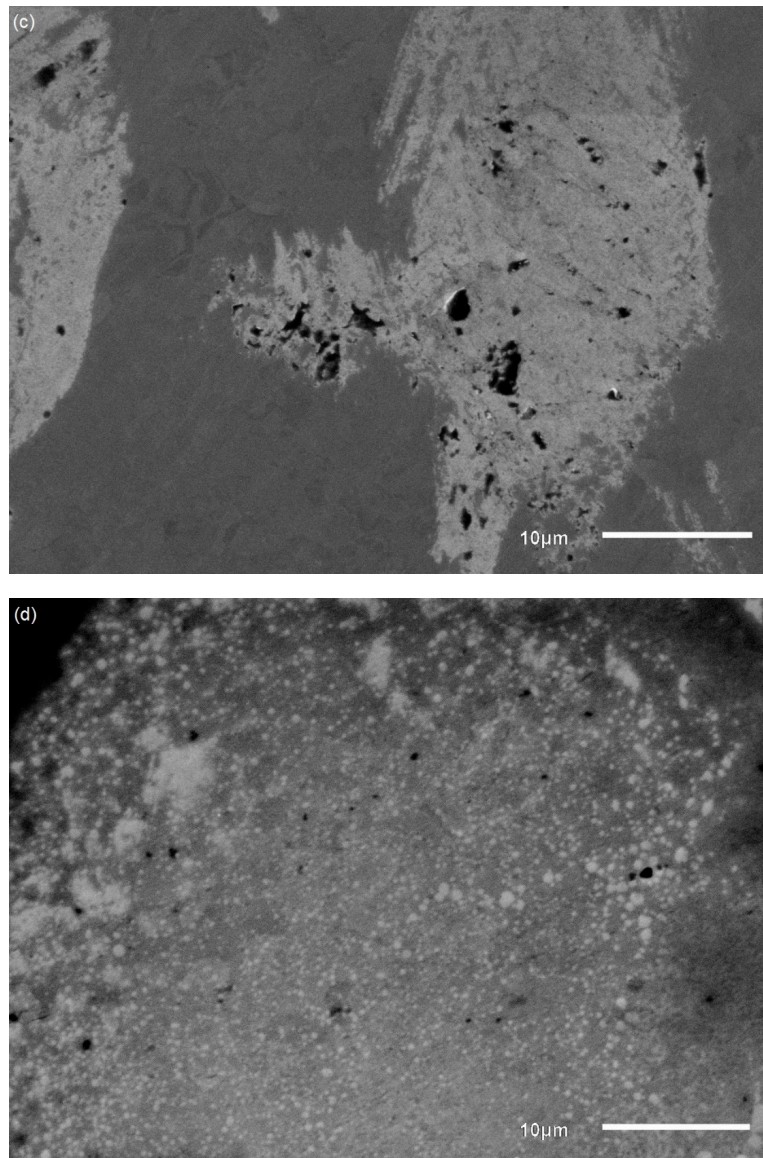

**Figure 11.** SEM images of coupons after biofilm removal after immersion in pH 7 (**a**); pH 8 (**b**); pH 9 (**c**); pH 10 (**d**) environments (black circular areas represent pit formation).

**Table 3.** Element concentrations (mass %) of coupons with biofilm immersed in different pH environments.

|       | C    | Si   | S    | Cr    | Fe    | Ni   |
|-------|------|------|------|-------|-------|------|
| pH 7  | 3.49 | 0.41 | 2.11 | 23.34 | 65.1  | 5.55 |
| pH 8  | 2.18 | 0.52 | 2.77 | 23.21 | 65.29 | 6.03 |
| pH 9  | 2.1  | 0.34 | 3.82 | 23.06 | 65.68 | 5.46 |
| pH 10 | 2    | 0.39 | 3.11 | 23.39 | 65.01 | 6.1  |

## 4. Discussion

### 4.1. Effect of Environment pH on Bacteria Growth and Bacteria Attachment

Sulphate reducing bacteria have been found in various environments, including high pH environment. Some studies have shown that SRB cannot survive at pH above 9 with suggestions of using alkaline solutions to control SRB activity [36]. However, subsequent studies have found the presence of SRB in high alkaline environments [39,42,50–52], possibly in association with biofilms which can harbor

microniches to protect micro-organisms from harsh environments [53–55]. It has been shown that pH as well as other gradients exist in a biofilm [50,51]. The presence of SRB in high alkaline environments can also be explained by the adaptation of bacteria to their environment [52]. The adaptive mechanisms can include: (i) increased proton capture and retention by elevating transporter (monovalent cation/proton antiporters) and enzyme (e.g., ATP synthase) activities, (ii) increased metabolic acid production through metabolic changes, and (iii) changes in cell surface properties [56–58].

Bacteria generally modify their surrounding pH through metabolic activities [56]. SRB metabolism which results in the production of hydrogen sulphide and can decrease the concentration of $OH^-$ which lowers the pH of the environment. In this research, SRB at pH 10 were active but only after an extended lag phase. Speciation of $H_2S$ in the aqueous phase is governed by the reversible ionization reaction shown below [31]:

$$H_2S_{aq} + OH^- \rightarrow HS_{aq}^- + H_2O \tag{3}$$

$$H_2S_{aq} + 2OH^- \rightarrow S_{aq}^{2-} + 2H_2O \tag{4}$$

Another product carbon dioxide ($CO_2$) was produced by the oxidation of lactate from environment (Equation (7)) could combine with $OH^-$ to form bicarbonate ($HCO_3^-$) and carbonate ($CO_3^{2-}$), thus decreased the environmental pH [1,6]. It has been shown that the dissolution of hydrogen sulphide was pH dependent [31]. The data provided by U.S. Center for Environmental Research Information shows that at pH 7, roughly 50% of hydrogen sulphide is dissolved in solution, while at pH 9, more than 99% of hydrogen sulphide is dissolved in solution [59]. Thus, at higher pH, the dissolution rate of hydrogen sulphide is higher [31,59] and this was supported in this study which showed higher concentrations of dissolved sulphide at pH 9 and 10 than at pH 7 and 8.

At pH 10, there was less biofilm formation on the material surface compared to in other pH environments, due to the significant lag phase, (Figures 4 and 8). At pH 7, 8 and 9, the extent of bacterial adhesion was nearly the same as each other and the biofilm thickness was around 75 μm (Figure 4).

### 4.2. Effect of pH on Microbial Corrosion Behaviour

This study found that pH affected SRB metabolism, which in turn affected the microbial corrosion of the material at different pH. For the first two days' exposure, as can be seen from Nyquist plots (Figure 6a,b), the impedance of the working electrode in pH 10 environment was the highest, even though it had the lowest bacteria attachment as presented in Figure 4 and in the change in double layer capacitance (Figure 8). This can be explained as follows:

- For the first 2 h of exposure, the increase in biofilm thickness was too low to act as a barrier to prevent diffusion of metal ions to the environment. Therefore, the impedance of materials depended on their passive film. It has been shown in previous literature that passive film has semiconducting properties [60–62]. The films are characterised by the presence of electrical barriers developed at the film–electrolyte interface and at the junction [61]. The donors or acceptors in semiconducting passive layers are defects, including cationic and anionic vacancies or cationic interstitials. These vacancies act as the dopants. For instance, cation interstitials imparting n-type character and cation vacancies yielding p-type character of semiconducting properties of passive film. Thus, the presence of these dopants prevent the diffusion of ion from metal substrate and the penetration of damaging anion such as $Cl^-$ from electrolyte [61,62]. As the pH of the environment increases, the density of dopants increases, thereby increasing the corrosion resistance [10], represented by higher radius impedance of working electrode in pH 10 solution than others (Figure 6a).
- There was a gradually enrichment of chromium content on the passive film layer as pH increased. Thus, the space charge layer increases with the chromium content [10] which results in higher pitting resistance.

The impedance of the electrodes continued increasing during 14 days of immersion as the biofilm grew presented as increasing impedance radius (Figure 7) and total resistance (Figure 8). The electrodes impedance decreased after 14 days exposure indicating that corrosion of the material started around this time.

Hydrogen sulphide is one of the strongest promoters of hydrogen entry into the steel surface [15,63] where the passive film can be destroyed. The formation of biogenic metal sulphide film as a reaction of metal ion and sulphide can temporarily lead to the protection of iron against corrosion [6]. Table 3 shows the presence of sulphide in the matrix of corrosion products and biofilm formed on coupon surfaces. This film acts as a barrier by impeding the diffusion of metal ion to environment. This can be seen in Figure 7 where the total resistance of all working electrodes in different pH environments lowered after 18 days exposure as the rate of resistance drop decreased. At higher pH, the dissolution rate of $H_2S$ is higher than at lower pH environment [31], and this resulted in higher dissolved biogenic sulphide which damaged to passive film of materials. This can be seen in Figure 2 where dissolved sulphide at pH 9 and 10 environments were higher than at pH 7 and 8 environments. Sulphides strongly depassivate stainless steels favouring development of pitting on the surface. Thereby, corrosion of materials was higher at pH 9 environment than at lower pH environments. From Figure 6c-g, it can be clearly seen that the radius of impedance of working electrode in pH 9 environment decreased with time exposure which indicates a reduction in corrosion protection. The working electrode at pH 10 environment also exhibited this characteristic. In this case, after the lag phase, the sulphide production sharply increased leading to the loss in corrosion resistance of the electrode. After 28 days of immersion, the impedance of the working electrode in pH 10 environment was lower than in pH 7 and 8.

### 4.3. Corrosion Mechanism

The proposed overall mechanism of *D. vulgaris* induced corrosion is shown in Figure 12. *D. vulgaris* reduce sulphate ion from the environment by enzyme through a serious of reactions [64–66]. At first, the enzyme ATP-sulfurylase also known as adenylylsulphate pyrophosphorylase or sulphate adenylyltransferase, which uses ATP and sulphate to create adenosine 5′-phosphosulfate (APS). APS is subsequently reduced to sulphite and release adenosine monophosphate (AMP). Sulphite is then further reduced to sulphide, while AMP is turned into adenosine diphosphate (ADP) using another molecule of ATP. The overall process, thus, involves an investment of two molecules of the energy carrier ATP, which must to be regained from the reduction [67]. The source of electron comes from anodic reactions including iron ion and oxidation reaction of lactate from environment. Thus, the redox reactions are as following [68,69]:

Anodic:

$$4Fe \rightarrow 4Fe^{2+} + 8e^- \quad E_o = -447 \text{ mV} \tag{5}$$

$$CH_3CHOHCOO^- + H_2O \rightarrow CH_3COO^- + CO_2 + 4H^+ + 4e^- \quad E_o = -430 \text{ mV} \tag{6}$$

Cathodic:

$$SO_4^{2-} + 9H^+ + 8e^- \rightarrow HS^- + 4H_2O \quad E_o = -217 \text{ mV} \tag{7}$$

Corrosion products formation:

$$Fe^{2+} + S^{2-} \rightarrow FeS \tag{8}$$

$$3Fe^{2+} + 6OH^- \rightarrow 3Fe(OH)_2 \tag{9}$$

$$Fe^{2+} + CO_2 + OH^- \rightarrow FeCO_3 + H^+ \tag{10}$$

The cathodic reaction was the consumption of electron by bacteria for their metabolism. The half cells potentials were calculated under standard condition of 25 °C, pH 7, 1 M concentration for solutes and 1 bar partial pressure for gases. Thus, the Gibbs energy between Equations (5) and (7), Equations (6)

and (7) were $-177.56$ kJ. mol$^{-1}$ and $-164.44$ kJ. mol$^{-1}$. The negative Gibbs energy suggests that the two redox reactions are highly favourable and corrosion of stainless steel can occur.

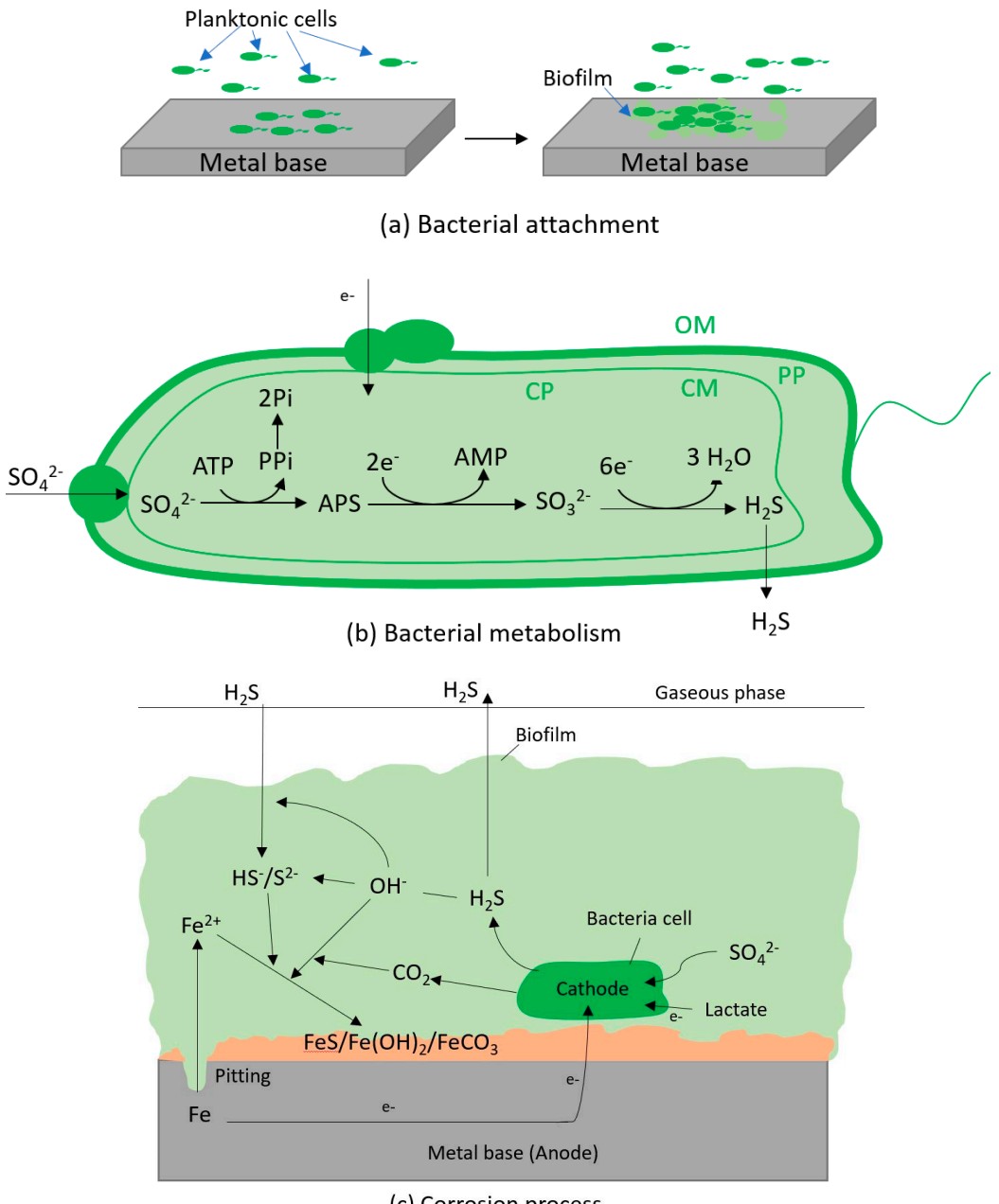

**Figure 12.** Schematic of changing pH mechanism and corrosion mechanism caused by SRB breakdown by two main steps including adhesion (**a**), bacterial metabolism (**b**) and corrosion process (**c**); bacterial metabolism including specific proteins in the outer membrane (OM), periplasm (PP) and cytoplasmic membrane (CM) allow electron go through for sulphate reduction process in the cytoplasm (CP); corrosion process including the production of hydrogen sulphide ($H_2S$) produced by bacterial metabolism which consumes electron from iron and lactate (nutrient) and the combination of iron ion with sulphide and hydroxide to form corrosion products ($FeS/Fe(OH)_2$).

The change in environmental pH is basically caused by the reaction of bacterial metabolites with hydroxide ion $OH^-$ from the environment, thus a reduction in the concentration of hydroxide ions. Another reason that is due to the precipitation reaction of hydroxide ion with cation released from materials which also caused in decrease the concentration of hydroxide ion. The biogenic sulphide

produced by bacteria metabolism caused the dissolution of the materials along with hydroxide from the environment, thus resulted in precipitation of metal sulphide and metal hydroxide as corrosion products. One of the important consequences of these electrochemical reactions is that the hydroxide ions are now part of the corrosion products and this results in lowering the pH by increasing the $H^+$ ions. This is clearly seen in this research as all the pH of the solutions reach a value of about 7.5. It appears that the pH regulation by bacteria is a common affair although more studies are needed in this area. Rahmi et al. [70] found that *Vibrio tapetis* also modified the pH of various laboratory media (6–8.7) to an average value of 7.71.

In abiotic environments, higher pH results in a lower corrosion rate compared to at lower pH due to the protection from more stable passivity [10,71,72]. In contrast, when SRB are present, greater production of biogenic sulphides/hydrogen sulphide at higher pH results in increased corrosion rates compared to lower pH. This research also showed SRB can survive and be active at pH 10 following an initial lag phase and modified the environmental pH to optimal pH. Therefore, in natural environments where SRB are present, the use of high $OH^-$ ion concentrations to control corrosion [72] should be reconsidered, and in fact could be detrimental in industrial settings [36,73].

## 5. Conclusions

Corrosion behaviour of DSS 2205 in alkaline marine environment was studied. The main conclusions were:

- *Desulfovibrio vulgaris* was active in alkaline environments at pH 7–9. At pH 10, there was an initial lag phase of around 8 days.
- The pH of the bulk environment decreased to pH 7.5 during exposure time due to the production of corrosion products.
- For the first two days of exposure, the corrosion resistance of the materials depended mostly on their passive film. Higher pH environments could support higher corrosion resistance of materials.
- After 28 days exposure to SRB environment, the pH of the environment shifted to pH 7.5 and corrosion still occurred. Thus, strategies to increase the pH to control MIC is unlikely to be successful.
- The corrosion rate of DSS 2205 at pH 9 was higher than other pH environment as it has the highest current density ($0.867 \text{ mA} \cdot \text{cm}^{-2}$) due to higher dissolution of hydrogen sulphide.
- A mechanism of the change in environmental pH and the corrosion of duplex stainless steels in the presence of SRB in alkaline environment is proposed.

**Author Contributions:** Conceptualisation, T.T.T.T.; Data curation, T.T.T.T.; Formal analysis, T.T.T.T.; Investigation, T.T.T.T.; Methodology, T.T.T.T.; Project administration, T.T.T.T. and K.K.; Supervision, K.K., A.P. and S.T.; Writing—original draft, T.T.T.T.; Writing—review & editing, T.T.T.T., K.K., A.P. and S.T. All authors have read and agreed to the published version of the manuscript.

**Funding:** This research was supported by an Australian Research Training Program Scholarship provided through Charles Darwin University; grant number is 1578028.

**Acknowledgments:** The authors would like to acknowledge INPEX Corporation, Japan for hosting Tien Tran through an internship at INPEX, Japan and INPEX Technical Research Centre staff for supporting the use of SEM-EDS and Steven Mason, School of Chemistry and Molecular Biosciences, The University of Queensland for imaging with ZEISS LSM 510 META confocal laser scanning microscope.

**Conflicts of Interest:** The authors declare no conflict of interest.

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
