# Peer review of "Effect of Alkaline Artificial Seawater Environment on the Corrosion Behaviour of Duplex Stainless Steel 2205"

_applsci, doi:10.3390/app10155043_

Round 1

Reviewer 1 Report

Reviewers' comments:

Manuscript number: applsci-866696

Title: Effect of alkaline artificial seawater environment on the corrosion behaviour of duplex stainless steel 2205

Comments: 

The manuscript reported on Effect of alkaline artificial seawater environment on the corrosion behaviour of duplex stainless steel 2205. The manuscript needs a detailed editing. It cannot be recommended for publication in the present form. I hope the following points would be helpful for the authors.

- The introduction section should be improved.

- Fig. 1 Three electrodes cell, is make clear.

- Several faults: are added or missing spaces between words: see PDF file.

- Authors should include OCP decay study.

- Give more detail for the determination of equivalent circuit model.

- Authors must but reference for each equation used.

- In part SEM: how the energy of the accelerator beam used?

- A new section detailing the corrosion mechanism should be provided. 

- Conclusions, the author should add some qualitative data of the results.

- References: there are recent references in 2019 and 2020 treating the same subject, you can use. And make all references in same format for volume number, page number and journal name.

So that I recommended this manuscript to major revision and for future process.

Reviewer 2 Report

Dear Authors and Editor,

In review I received a manuscript entitled “Effect of alkaline artificial seawater environment on the corrosion behaviour of duplex stainless steel 2205” considered for publication in MDPI journal “Applied Sciences”.

The manuscript reports on SRB corrosion behavior on 2205 SS, which is of interest for practical application of structural materials in different industries. The research is well-planned and conducted. I think it is appropriate for publication, and here are some of my comments:

  • Nomenclature: EDX. As a method, the energy dispersive X-ray spectroscopy can be abbreviated either EDS or EDXS, I suggest using one of these options. For instance, in P17|L410 SEM-EDS abbreviation is used.
  • Fig 1 subtitle needs a proper description of what we see on the figure. Think of the subtitle as a short abstract of what we should learn from the image itself.
  • P3|L86: (EDX-8100). MDPI Journals usually demands a method name (model, producer, country) naming of the used equipment. Also goes for P3|L123.
  • P4|155 is referring to EDX (EDXS), which is probably mounted on SEM. Also the “main” machine should be mentioned somewhere, especially when spectrometer is only an extension of it. Proper operating conditions for the EDXS analysis should be stated (e.g. acc. voltage).
  • P4|159 SEM is mentioned, so the abbreviation should be explained first (scanning electron microscope) and conditions of operation provided, together with any sample preparation involved (e.g. C or Au coating, bacteria stabilization process, etc.).
  • P17|291: SEM micrographs need explanation on mode used (like secondary electrons, SE), and a brief description of what we see on the images (how the contrast is interpreted). You can even use additional features on the micrographs itself, like arrows or text, to help the reader to identify the pits and voids.
  • P18|297: “… more pits … that the others”. We are absent of the reference material (the “before” sample), so any suggestions on how to define these pits are the effect of corrosion? SEM images show only limited size of the surface, hence a bit of more description would be appropriate here. Like, using automotive methods in ImageJ to quantify voids per area, and briefly compare them and correlate to results of higher corrosion rate at higher pH?
  • Was the S detected on the surface also by EDS, or just by ICP-MS? If yes, the conditions of SEM-EDS analysis have to be stated, together with possible standards used and how the results were interpreted. As mentioning Fe-S phases, was any of them detected (observed) or this is just speculation?

The text needs additional (light) proofreading. Nothing serious, but just a polish that would benefit the reader.

Additionally, I would suggest the Authors to slightly modify the type of presentation of the results. In the current version, paragraphs start with “Fig. X shows…”, hence it reads like an extended figure subtitle. Take this as a suggestion, not criticism, and would definitely benefit to the broader audience.  Otherwise, I have no further objections against the publication of the manuscript. 

Round 2

Reviewer 1 Report

Reviewers' comments:

The authors revised the manuscript according to the reviewers' comments.

So that I recommended this manuscript accept for publication in Applied Sciences.

Reviewer 2 Report

I have no further objections regarding the publication.